# 8-Hydroxydaidzein, an Isoflavone from Fermented Soybean, Induces Autophagy, Apoptosis, Differentiation, and Degradation of Oncoprotein BCR-ABL in K562 Cells

**DOI:** 10.3390/biomedicines8110506

**Published:** 2020-11-16

**Authors:** Pei-Shan Wu, Jui-Hung Yen, Chih-Yang Wang, Pei-Yi Chen, Jui-Hsiang Hung, Ming-Jiuan Wu

**Affiliations:** 1Department of Pharmacy, Chia Nan University of Pharmacy and Science, Tainan 717, Taiwan; dc7575@gmail.com; 2Department of Molecular Biology and Human Genetics, Tzu Chi University, Hualien 970, Taiwan; imyenjh@mail.tcu.edu.tw (J.-H.Y.); pyc571@gmail.com (P.-Y.C.); 3Institute of Medical Sciences, Tzu Chi University, Hualien 970, Taiwan; 4Ph.D. Program for Cancer Molecular Biology and Drug Discovery, Taipei Medical University, Taipei 11031, Taiwan; chihyang@tmu.edu.tw; 5Graduate Institute of Cancer Biology and Drug Discovery, Taipei Medical University, Taipei 11031, Taiwan; 6Center of Medical Genetics, Buddhist Tzu Chi General Hospital, Hualien 970, Taiwan; 7Department of Biotechnology, Chia Nan University of Pharmacy and Science, Tainan 717, Taiwan; hung86@mail.cnu.edu.tw

**Keywords:** K562, 8-hydroxydaidzein, apoptosis, autophagy, BCR-ABL, MAPK, NF-κB

## Abstract

8-Hydroxydaidzein (8-OHD, 7,8,4′-trihydoxyisoflavone) is a hydroxylated derivative of daidzein isolated from fermented soybean products. The aim of this study is to investigate the anti-proliferative effects and the underlying mechanisms of 8-OHD in K562 human chronic myeloid leukemia (CML) cells. We found that 8-OHD induced reactive oxygen species (ROS) overproduction and cell cycle arrest at the S phase by upregulating p21^Cip1^ and downregulating cyclin D2 (CCND2) and cyclin-dependent kinase 6 (CDK6) expression. 8-OHD also induced autophagy, caspase-7-dependent apoptosis, and the degradation of BCR-ABL oncoprotein. 8-OHD promoted Early Growth Response 1 (EGR1)-mediated megakaryocytic differentiation as an increased expression of marker genes, *CD61* and *CD42b*, and the formation of multi-lobulated nuclei in enlarged K562 cells. A microarray-based transcriptome analysis revealed a total of 3174 differentially expressed genes (DEGs) after 8-OHD (100 μM) treatment for 48 h. Bioinformatics analysis of DEGs showed that hemopoiesis, cell cycle regulation, nuclear factor-κB (NF-κB), and mitogen-activated protein kinase (MAPK) and Janus kinase/signal transducers and activators of transcription (JAK-STAT)-mediated apoptosis/anti-apoptosis networks were significantly regulated by 8-OHD. Western blot analysis confirmed that 8-OHD significantly induced the activation of MAPK and NF-κB signaling pathways, both of which may be responsible, at least in part, for the stimulation of apoptosis, autophagy, and differentiation in K562 cells. This is the first report on the anti-CML effects of 8-OHD and the combination of experimental and in silico analyses could provide a better understanding for the development of 8-OHD on CML therapy.

## 1. Introduction

Chronic myeloid leukemia (CML) is a hematopoietic disease characterized by the expression of the BCR-ABL fusion oncoprotein, which is caused by a reciprocal translocation between chromosomes 9 and 22 [t(9;22)(q34;q11)] [1]. BCR-ABL fusion protein exerts a deregulated tyrosine kinase activity that activates proliferative and anti-apoptotic signaling pathways and leads to the malignant expansion of pluripotent stem cells in bone marrow [2]. Historically, CML was treated with hematopoietic stem cell transplantation, chemotherapy, and interferon-α [3]. Small molecule tyrosine kinase inhibitors (TKIs), such as imatinib, bosutinib, dasatinib, nilotinib, radotinib, and ponatinib, have been developed to treat CML by blocking the kinase domain of the BCR-ABL oncoprotein. However, a variety of adverse effects, such as off-target and metabolic toxicities, resistance to TKI therapy due to BCR-ABL mutations, and progression to advanced disease, render the need to pursue novel therapeutic strategies [3,4].

Since the BCR-ABL protein plays a crucial role in the pathogenesis of CML, its degradation is a new strategy for overcoming the problem of TKI resistance [5,6]. Targeting approaches include ubiquitin–proteasome, ubiquitin–lysosome, autophagy–lysosome, and caspase-mediated degradation pathways [5]. For example, arsenic sulfide As_4_S_4_ and diterpenoid oridonin promote ubiquitin–proteasome degradation of BCR-ABL [7,8]. Andrographolide, geldanamycin (GA), and 17-allylamino-17-demethoxygeldanamycin (17-AAG) downregulate the BCR-ABL oncoprotein by inducing HSP90 cleavage [9,10]. Arsenic trioxide (As_2_O_3_) promotes p62/SQSTM1-mediated autophagic degradation of the BCR-ABL [11]. Platinum pyrithione and xanthohumol downregulate BCR-ABL level through caspase-mediated degradation [12,13].

Isoflavones are dietary phytoestrogens mainly produced by Fabaceae family [14]. They are able to compete with estrogen hormone, 17β-estradiol, for the ligand-binding domain of the estrogen receptors and are used in the prevention and/or treatment of cancers, cardiovascular disease, osteoporosis, and postmenopausal symptoms [15,16,17]. Furthermore, genistein can inhibit tyrosine kinase activity and displays cancer chemopreventive activity [18]. 8-Hydroxydaidzein (8-OHD, 7,8,4′-trihydroxyisoflavone, NSC-678112) is a hydroxylated derivative of daidzein commonly isolated from fermented soybean products. It has recently attracted research attention due to its strong anti-inflammatory, antioxidant, antitumor, anti-melanogenesis, and hepatoprotective activities [19,20,21,22,23,24]. It was found that 8-OHD had strong anti-proliferative activity toward HL-60 human promyelocytic leukemia cells [25].

In this study, we evaluated the anti-CML effects of 8-OHD with a focus on cell cycle arrest, apoptosis, autophagy, differentiation, and BCR-ABL levels in K562 cells, which is a model cell line derived from a female CML patient in blast crisis [26]. High-throughput microarray analysis is a valuable tool for early-stage drug discovery decision-making due to its potential to detect adverse effects at the transcriptomic level [27]. Thus, microarray analysis was employed to explore the differentially expressed genes (DEGs) in 8-OHD-treated K562 cells. Then, Gene Ontology (GO), (Kyoto Encyclopedia of Genes and Genomes) KEGG, and BioCarta pathways were analyzed [28,29,30]. We further established a protein–protein interaction (PPI) network of DEGs to screen out the hub genes with a high degree of connectivity in the biological process.

Our data showed that 8-OHD induces reactive oxygen species (ROS) overproduction and cell cycle arrest at the S phase by upregulating p21^Cip1^ and downregulating cyclin-dependent kinase 6 (CDK6) and cyclin D2 (CCND2) expression. Caspase-7-dependent apoptosis and p62/UVRAV-dependent autophagy may be responsible for the decreased level of BCR-ABL. Furthermore, 8-OHD promoted Early Growth Response 1 (EGR1)-mediated megakaryocytic differentiation. 8-OHD also activated mitogen-activated protein kinase (MAPK) and nuclear factor-κB (NF-κB) signaling pathways, and both may be responsible for the stimulation of apoptosis, autophagy, and differentiation in K562 cells. This is the first transcriptomics report on the anti-leukemic effects of 8-OHD, and the obtained results can provide a more thorough understanding about potential development of 8-OHD in CML therapy.

## 2. Experimental Section

### 2.1. Preparation of 8-Hydroxydaidzein (8-OHD, 7,8,4′-trihydoxyisoflavone, NSC-678112)

8-OHD was isolated from soybean fermented by *Aspergillus oryzae*, and the NMR spectral data and purification of 8-OHD was reported previously [20,21,24]. The chemical structure of 8-OHD was shown in Figure 1a.

### 2.2. Cell Culture

The K562 cells were provided by the Bioresource Collection and Research Center (Hsinchu, Taiwan) and cultured in RPMI-1640 medium supplemented with 10% fetal bovine serum (FBS), 1% nonessential amino acids (NEAA), 100 units/mL of penicillin, and 100 µg/mL of streptomycin (Thermo Fisher Scientific, Inc., Rockford, IL, USA) in a 5% CO_2_ incubator at 37 °C.

### 2.3. Cell Proliferation and Viability Analysis

K562 cells (5 × 10^5^/mL) were treated with vehicle (0.1% DMSO) or 8-OHD (12.5–100 μM) for 24 h or 48 h. Cell viability was analyzed by adding 1/10 volume of 0.5% MTT (3-(4,5-dimethylthiazol-2-yl)-2,5-diphenyltetrazolium bromide, in PBS) and incubating for another 3 h. Then, MTT solution was removed by centrifugation, and the formazan crystals produced inside cells were dissolved by DMSO, and the absorbance at 550 nm was measured spectrophotometrically [31]. The number of viable cells after treatment was further accessed by the trypan blue exclusion test as described in the literature [32].

### 2.4. Cell Cycle Analysis

K562 cells were synchronized by serum starvation overnight prior to shifting cells to 8-OHD-containing normal medium for 24 h. Then, cells were washed twice with PBS and fixed in ice-cold 70% ethanol overnight. The fixed cells were stained with 1 mL DNA-staining buffer (20 µg/mL of propidium iodide and 50 µg/mL of RNase in PBS) in the dark at 4 °C for 15 min before flow cytometry analysis (FACScan, BD Biosciences, San Jose, CA, USA). The singlet cell population was gated on the dot plot of FL2-A vs. FL2-W to exclude cell debris and aggregates. To evaluate the cell cycle, FL2-A histogram of gated population was analyzed by FlowJo software (FlowJo v7.6, LLC, Ashland, OR, USA) with Dean-Jett-Fox model.

### 2.5. Intracellular Reactive Oxygen Species (ROS) Assay

The fluorescence probe 2′,7′-dichlorodihydrofluorescein diacetate (H_2_DCFDA) (Invitrogen) was used to measure ROS production. K562 cells were incubated with 8-OHDA (25–100 μM) for 24 h. Cells were collected by centrifugation and then washed with PBS. Subsequently, cells were loaded with 200 μL of 10 µM H_2_DCFDA under dark for 30 min. Then, cells were washed twice with cold PBS and analyzed by fluorometer at Ex/Em: 495/530 nm. The relative ROS production from 10,000 cells was determined as the percentage of control after background subtraction [33].

### 2.6. Western Blot Analysis

Total cell lysate was prepared from cultured K562 cells using radioimmunoprecipitation assay buffer (RIPA buffer), while nuclear extracts were by a nuclear extraction kit (Cayman Chemical, Ann Arbor, Michigan, USA). Then, Bradford assay was employed to measure the protein concentration (Bio-Rad Laboratories, Hercules, CA, USA).

Equal amounts of protein were subjected to separate on 5–12% SDS-PAGE. Following electrophoretic separation, the proteins were transferred to a polyvinylidene difluoride (PVDF) membrane and then were blocked with freshly made buffer (5% skim milk in PBS with 0.05% Tween 20, pH 7.4). Then, the membrane was probed with specific primary antibody (Table 1) overnight at 4 °C. After rinsing, horseradish peroxidase (HRP)-conjugated secondary antibody (Jackson ImmunoResearch, West Grove, PA, USA) was then added and incubated for 1 h. The antigen–antibody reaction was detected using enhanced chemiluminescence detection (GE Healthcare, Wauwatosa, WI, USA).

### 2.7. RNA Extraction and Reverse Transcription Quantitative PCR

RNA was extracted from K562 cells with the Illustra RNA Spin Mini RNA Isolation Kit (GE Healthcare, Wauwatosa, WI, USA). High-Capacity cDNA Archive kit (Thermo Fisher Scientific, Inc., Rockford, IL, USA) was used for cDNA synthesis. Quantitative PCR was performed using a Power SYBR Green PCR Master Mix (Thermo Fisher Scientific, Inc., Rockford, IL, USA) in a total volume of 20 μL that contained 0.4 μM of each primer (Table 2). PCR program consisted of a pre-incubation for 2 min at 95 °C, followed by 40 cycles at 94 °C for 15 s and 60 °C for 60 s (ABI StepOne Real Time PCR System). Specificity verification was done by the melting curve. The relative mRNA expression was normalized with β-actin expression and then calculated by comparative Ct method.

### 2.8. Analysis of Cell Morphology

K562 cells were incubated with vehicle (0.1% DMSO) or 8-OHDA (100 μM) for 24 and 48 h. Suspension cells (2.5 × 10^4^) were added on Polysine™ slides and centrifuged at 200 rpm for 5 min using Cytospin™ 4 Cytocentrifuge (Thermo Fisher Scientific, Inc., Rockford, IL, USA). Slides were carefully removed and air-dried prior to staining with Wright-Giemsa dye (MilliporeSigma, St. Louis, MO, USA). The cell morphology and lobulation of the nucleus were analyzed as described previously [34].

### 2.9. Microarray Analysis

Cells were treated with vehicle (0.1% DMSO) or 8-OHD (100 μM) for 48 h. Microarray analysis was performed as previously described [35,36]. Briefly, fluorescent antisense RNA (aRNA) targets were prepared from 1 μg total RNA samples using Onearray Amino Allyl aRNA Amplification Kit (Phalanx Biotech Group, Hsinchu, Taiwan) and Cy5 dyes (Amersham Pharmacia, Piscataway, NJ, USA). Fluorescent targets were hybridized to the Human One Array Plus Version 7.1 (HOA 7.1, Phalanx) with phalanx hybridization buffer using Phalanx OneArray Plus Protocol. The signals of interest were scanned via an Agilent G2505C scanner using Agilent 0.1 XDR Protocol (Agilent Technologies, Santa Clara, CA, USA). The fluorescence intensity of each spot was analyzed and processed by GenePix 4.1 software (Molecular Devices, Sunnyvale, CA, USA) and a Rosetta Resolver System (Rosetta Biosoftware, Seattle, WA, USA), respectively. Differentially expressed genes (DEGs) related with the treatment of 8-OHD were screened, and those with fold change values larger than 2 or less than −2 and *p* < 0.05 were selected.

### 2.10. Gene Ontology, KEGG, and Biocarta Pathways and Protein–Protein Interaction Analysis

Gene Ontology (GO) term analysis [28] as well as KEGG [30] and Biocarta [29] Pathways of DEGs were further analyzed using the Database for Annotation, Visualization, and Integrated Discovery (DAVID, http://david.ncifcrf.gov) (version 6.8), an online biological information database, and *p* < 0.05 was used as the cut-off criterion [37]. The protein–protein interactions were analyzed using STRING version 11 [38] on 2 October 2020 (https://string-db.org/).

### 2.11. Pathway Enrichment and Process Network Analysis

Molecular functions and the pathway of Gene Ontology (GO) term in MetaCore (GeneGo, Inc., St. Joseph, MI, USA) were further used to screen and analyze the signaling pathway and process networks modulated by 8-OHD. The REVIGO web-based tool was used to summarize and remove redundant GO terms. Gene Set Enrichment Analysis (GSEA) [39] was used to determine the statistically significant molecular pathway with 8-OHD treatment.

### 2.12. Statistical Analysis

All experiments were repeated at least three times, and the values were expressed as the mean ± SD. The results were analyzed using one-way ANOVA with Dunnett’s post hoc test, and a *p* value < 0.05 was considered statistically significant.

## 3. Results and Discussion

### 3.1. 8-OHD Reduces the Proliferation and Viability of K562 Cells

The K562 cell line is a well-known model of human chronic myeloid leukemia (CML). We first examined the effect of 8-OHD on K562 cell proliferation by MTT assay. Figure 1b shows that 8-OHD (12.5–100 μM) significantly decreased cell growth in a dose- and time-dependent manner. Cell proliferation decreased to 68.7% and 56.8% of vehicle control, 24 and 48 h after 100 μM 8-OHD treatment, respectively. It has been shown that MTT assay sensitivity decreases as cell concentration increases and causes an overestimation of viability [40]. Thus, we reconfirmed the cell viability by trypan blue exclusion assay. Figure 1c shows that 8-OHD exerted inhibitory effects on K562 cell viability in a dose- and time-dependent manner. The IC_50_ (half maximal cell viability inhibitory concentration) were 91.8 μM at 24 h and 49.4 μM at 48 h. Our results were in accordance with previous studies, which show that 8-OHD at high concentrations, 40 μM and 100 μM, exerted more than 40% cytotoxicity in B16 murine melanoma and Caco-2 cells, respectively [41,42].

### 3.2. 8-OHD Causes Cell Cycle Arrest at S phase

We further investigated how 8-OHD affects the cell population distribution in the cell cycle by staining the cellular DNA with propidium iodide (PI) in K562 cells. As shown in Figure 2a–c, 8-OHD caused a significant increase in S phase cell population, from about 32.2% of vehicle control to about 37.3% of 50 μM 8-OHD (*p* < 0.05) and 44.4% of 100 μM 8-OHD (*p* < 0.01). This was accompanied with decreases in the proportion of cells in the G_2_/M phase of the cell cycle. Since cell proliferation is inhibited by 8-OHD (Figure 1), it seems unlikely that the increase in the S phase represents an increase in cells that are actively replicating DNA [43]. This rather indicates that K562 cells are arrested in the S phase upon treatment with 8-OHD (50–100 μM).

It has been shown that overexpression of the CDK inhibitor p21^Cip1^ (CDKN1A, also known as p21^WAF1^) induces S-phase arrest [43,44]. Figure 2d shows that the p21^Cip1^ protein was upregulated by 8-OHD (25–100 μM) in a dose-dependent manner. It has been reported that p21^Cip1^ is an inhibitor of the cyclin D-CDK complexes [45]. Thus, we further examined the expression of cyclin-dependent kinase 6 (CDK6) and cyclin D2 (CCND2) by Western blot analysis. As shown in Figure 2c, CDK6 and cyclin D2 were dose-dependently inhibited by 8-OHD (25–100 μM). Thus, 8-OHD induced K562 cell cycle arrest at the S phase by upregulating p21^Cip1^ so as to downregulate the expression of CDK6 and cyclin D2.

### 3.3. 8-OHD Causes ROS Production and Apoptosis in K562 Cells

It was found previously that 8-OHD induced ROS production and caused cell death in Caco-2 cells [42]. Therefore, we accessed the intracellular ROS production of K562 cells. Figure 3a shows that 8-OHD (25–100 μM) significantly increased ROS production in a dose-dependent manner in K562 cells, and 8-OHD (100 μM) enhanced about 400% ROS production as compared with vehicle control after 24 h treatment (*p* < 0.01).

We further investigated whether K562 cells undergo apoptosis in response to 8-OHD. Caspase-3 (CASP3) and -7 (CASP7) are the most important executioner caspases involved in apoptotic cells [46]. Figure 3b shows that the treatment of K562 cells with 8-OHD (100 μM) for 24 h induced an increase in the expression and cleavage of caspase-7, which was more prominent after 48 h treatment (Figure 3c). In contrast, 8-OHD did not change the expression or activation of caspase-3 in K562 cells. It has been reported that caspase-3 and -7 are at the center of a fundamentally recursive apoptotic program, where they act as redundant signal amplifiers that are critical for activation of upstream and downstream apoptotic processes [47]. Our data indicate that caspase-7-dependent and caspase-3-independent apoptotic pathway is induced by 8-OHD.

Poly (ADP-ribose) polymerase-1 (PARP1) is the cellular substrate of caspases [48], caspase-7 is the most efficient caspase in PARP1 cleavage [49], and the proteolytic inactivation of PARP1 by caspases ensures the execution of the apoptotic process [50]. Figure 3b, c show that in parallel to caspase-7 activation, treatment of K562 cells with 100 μM 8-OHD for 24 and 48 h induced an increase in PARP1 cleavage.

It is well-known that natural compounds, such as vitamins and polyphenols, may play dual roles, antioxidant and pro-oxidant, in the prevention of cancer formation and cancer treatment [51,52,53]. It has been shown that 8-OHD, at lower μM concentrations, had strong free radical scavenging activities and decreased ROS production in cell-free and BV2 microglial cells [19,20,24,25]. On the other hand, 8-OHD could stimulate oxidative stress and caused Caco-2 cell death at higher concentrations [42]. Here, we reported that 8-OHD (50 and 100 μM) could stimulate ROS overproduction, which may cause caspase-7-dependent apoptosis in K562 cells.

### 3.4. 8-OHD Causes Autophagy and Decreases BCR-ABL Oncoprotein in K562 Cells

LC3s (MAP1LC3A, B, and C), the autophagosomal membrane proteins, are the most widely used markers of autophagosomes [54]. The conversion of LC3B from LC3BI (free form, ≈18 kDa) to LC3BII (phosphatidylethanolamine-conjugated form, ≈ 16 kDa) is an initiating step in autophagy in mammals [55]. Figure 4a,b show a dose-dependent increase in the LC3BII 24 and 48 h after 8-OHD (25–100 μM) treatment in K562 cells.

BCR-ABL tyrosine kinase is responsible for the proliferation and malignant transformation of CML. Therefore, we next investigated whether 8-OHD inhibits cell proliferation through decreasing the BCR-ABL protein level in K562 cells. As shown in Figure 4a, 8-OHD (100 μM) exerts a marked inhibitory effect on BCR-ABL expression in K562 cells after 24 h treatment. Furthermore, BCR-ABL levels were significantly attenuated by lower concentrations of 8-OHD (25–100 μM) after 48 h treatment (Figure 4b).

It was approved that ubiquitin-associated protein p62 (SQSTM1, autophagy receptor sequestosome 1), which binds to LC3, is an autophagy marker [56,57]. Increased expression of p62 (SQSTM1) was found in phorbol ester PMA- and the p38 MAPK inhibitor SB202190-induced autophagy in K562 cells [58]. Figure 4c shows that 8-OHD (25–100 μM) also stimulated *p62* (*SQSTM1*) mRNA upregulation in a dose- and time-dependent manner.

It has been reported that BCR-ABL protein degradation can be mediated via autophagy–lysosome pathway or caspase activation [5,11,59]. The above data indicate that 8-OHD causes cell death through caspase-7-dependent apoptosis and autophagy pathways, and both pathways may be responsible for, at least in part, the attenuation of the BCR-ABL protein level in K562 cells.

### 3.5. 8-OHD Promotes EGR1-Induced K562 Cell Differentiation

K562 cells represent stem cell precursors of the myeloid lineage and can differentiate into megakaryocyte/erythroid or granulocyte/macrophage lineages after exposure to various inducers [60,61]. It has been reported that autophagy is required for myeloid differentiation [58,62]. Furthermore, *p62* (*SQSTM1*) upregulation was found during megakaryocyte differentiation of K562 cells [58]. In addition, the BCR-ABL protein has been shown to confer on cells the characteristics of differentiation inhibition [63]. Since 8-OHD induced autophagy and *p62* expression, as well as decreased BCR-ABL protein level, we further investigated its effects on the expression of genes associated with megakaryocyte/erythroid differentiation. As shown in Figure 5a, 8-OHD (25–100 μM) significantly increased the mRNA expression of *CD61* (integrin β3, *ITGB3*), a megakaryocytic marker gene, 24 and 48 h after treatment. *CD61* mRNA expression was increased by 5.0-, 13.5- and 21.5-fold, 48 h after 25, 50, and 100 μM 8-OHD treatment (*p <* 0.01), respectively.

8-OHD (100 μM) also induced *CD42b* (Glycoprotein Ib Platelet Subunit Alpha, *GP1BA*) expression by 2.6- and 3.1-fold, 24 and 48 h after treatment in K562 cells, respectively (*p <* 0.01) (Figure 5b). Early growth response 1 (EGR1) is a transcriptional regulator that is involved in promoting the differentiation of K562 cells and suppressing leukemia tumorigenesis [34,64,65]. 8-OHD (25–100 μM) exerted a dose- and time-dependent induction effect on *EGR1* mRNA expression (Figure 5c). 8-OHD (100 μM) upregulated *EGR1* mRNA by 6.3- and 9.7-fold as compared with vehicle control, 24 and 48 h after treatment, respectively (*p <* 0.01).

It has been reported that multi-lobulation of the nucleus is a unique morphologic change during K562 megakaryocytic differentiation [34]. Figure 5d shows 100 μM 8-OHD promoted the formation of large cells with multi-lobulated nuclei 48 h after treatment. Microscope examination found that the ratio of cells with multi-lobulated nuclei increased from 0.83 ± 0.06% of untreated to 1.04 ± 0.13% and 3.75 ± 0.20% after 24 and 48 h 8-OHD (100 μM) treatment, respectively. This result indicates that a high concentration of 8-OHD (100 μM) induced K562 differentiation.

### 3.6. Analysis of 8-OHD-Modulated Gene Expression

From the above data, we discovered that a high concentration of 8-OHD (100 μM) induces apoptosis, autophagy, cell differentiation, and BCR-ABL downregulation in K562 cells. Recently, transcriptome analysis provides an ideal method to investigate molecular changes in response to stress or drug and is a valuable tool in early-stage drug discovery decision making [27]. Thus, we further investigated gene expression in 8-OHD-treated cells by human genome-wide microarray analysis. K562 cells were treated with vehicle or 8-OHD (100 μM) for 48 h, and transcriptome profiles were analyzed using the Human OneArray system, which contained 25,765 known genes. The general profile of transcriptome change is illustrated as a volcano plot (Figure 6a). Those with fold change values larger than 2 or less than −2, and *p* < 0.05 were selected as differentially expressed genes (DEGs). Among these, 1522 were upregulated and 1652 were downregulated by 100 μM 8-OHD. The most significantly down-related gene is *CRIP3* (*p* = 2.7 × 10^−7^), encoding a cysteine-rich protein with little known function. The most significantly upregulated gene is *SGMS2* (*p* = 1.0 × 10^−7^), encoding an enzyme sphingomyelin synthase 2, which is a regulator of cell surface levels of ceramide: an important mediator of signal transduction and apoptosis [66].

The 30 DEGs with the most fold change (15 up- and 15 downregulated) are presented in Appendix A. The most dramatic downregulated *HECW2* (downregulated by 64-fold) is a member of a family of E3 ubiquitin ligases, which play an important role in the proliferation, migration and differentiation. It has been reported that *HECW2* was one of the most downregulated genes by dasatinib, which is a selective tyrosine kinase receptor inhibitor that is used in the therapy of chronic myelogenous leukemia (CML) [67]. RT-Q-PCR in Figure 6b reconfirmed that 8-OHD (50 and 100 μM) repressed *HECW2* transcription to 0.075- and 0.011-fold of vehicle, respectively, 48 h after treatment (*p <* 0.01).

Microarray data also show that *CD69* expression was strongly downregulated by 8-OHD (down to 0.07-fold of vehicle). CD69, a member of the calcium-dependent lectin superfamily of type II transmembrane receptors, was reported as the marker for BCR-ABL activity in chronic myeloid leukemia [67]. In addition, CD69 was found to inhibit the apoptosis and differentiation in K562 cells [68]. RT-Q-PCR in Figure 6b validated that 8-OHD (50 and 100 μM) inhibits *CD69* mRNA expression after 48 h treatment. The above data support the notion that 8-OHD decreases BCR-ABL levels, which may cause *CD69* downregulation and in turn induce apoptosis and differentiation in K562 cells.

We also found that *LOXL2*, which encodes lysyl oxidase-like protein 2, was upregulated by more than 180-fold after 48 h treatment with 8-OHD (100 μM) (Figure 6a and Appendix A). Both the down- and upregulation of *LOX2* in tumor tissues and cancer cell lines has been reported in the literature, suggesting a paradoxical role for *LOX2* as a tumor suppressor and a metastasis promoter gene [69]. The exact role of *LOXL2* overexpression on cell proliferation/differentiation in 8-OHD-treated K562 cells remains unclear at this stage.

### 3.7. Analysis of 8-OHD-Modulated Biological Processes

To determine which Gene Ontology (GO) term was modulated by 8-OHD (100 μM), we performed an analysis of the enrichment in the relevant ontological groups from the GO database. A whole set of DEGs consisting of 3174 genes was subjected to functional annotation and clusterization using the Database for Annotation, Visualization, and Integrated Discovery (DAVID) bioinformatics tools.

Appendix A presented the only GO BP (biological process) groups that fulfill the following criteria: *p* < 0.05 and the minimal number of genes per group ≥ 30. REVIGO was used for the visualization of a non-redundant GO term set in multiple ways to assist in interpretation. The scatterplot shows the cluster representatives in a two-dimensional space derived by applying multidimensional scaling to a matrix of the GO terms’ semantic similarities [70]. It was found that small GTPase-mediated signal transduction, autophagy, transcription from RNA polymerase II promoter, and protein phosphorylation processes were significantly changed in the 8-OHD-treated cells (Figure 7a).

Furthermore, Over Representation Analysis (ORA) is a widely used approach to determine whether known biological functions or processes are enriched in DEGs, and GO-elite was a tool for ontology pruning [71]. It was found that biological processes related to oxidative stress, signal transduction, autophagy, cell death, and differentiation appeared significantly associated with upregulated DEGs in 8-OHD-treated groups. On the other hand, oxidative phosphorylation, cellular catabolic, and primary metabolic processes were significantly associated with downregulated DEGs in 8-OHD-treated groups (Figure 7b). These results were in good agreement with our experimental data that 8-OHD stimulates ROS, cell death, differentiation and changes of BCR-ABL tyrosine kinase signaling, as well as decreases normal cellular metabolism.

To uncover dependencies of 8-OHD-treated apoptosis with DEGs, enrichment for genes in the signature “Hallmark_Apoptosis” using GSEA was employed [72]. We found that most of the genes associated with the signature “Hallmark_Apoptosis” were stimulated by 8-OHD (Figure 7c).

Table 3 shows 35 DEGs were associated with autophagy (GO: 0006914). Among them, 29 genes were upregulated, and six genes are downregulated by 100 μM 8-OHD. We further exploit protein interaction for those upregulated genes in this category by String analysis. Markov clustering algorithm (MCL) with inflation factor = 2 was employed to find cluster structure in graphs by a mathematical bootstrapping procedure [73]. As shown in Figure 7d, two clusters in the protein–protein interaction (PPI) network. One cluster with light green color proteins are mainly involved in intracellular signal transduction and vesicular trafficking process during autophagy. The other cluster with red color includes highly induced genes such as p62 (SQSTM1), autophagy-related proteins (ATG13 and ATG2A), and UVRAG. Among these, UVRAG has 10 edges within the cluster and is considered as a hub. UVRAG has been suggested to be involved in the maturation of autophagosomes and endocytosis during autophagy [74,75]. UVRAG upregulation also participates in the p62-mediated autophagic degradation of BCR/ABL in CML cells [76].

p62 (SQSTM1) has 10 edges (seven connect to the same cluster, and three connect to different clusters) and is considered as another hub. It has been reported that p62 (SQSTM1) inactivates the pro-oncogenic signaling through interaction with BCR-ABL and the subsequent autophagic degradation of BCR-ABL in CML [11]. In conclusion, the bioinformatics and Western blot (Figure 4) analyses suggest that 8-OHD decreased BCR-ABL possibly via autophagic degradation post-translationally.

### 3.8. Analysis of Signaling Pathways and Network Processes Using KEGG, BioCarta, and MetaCore

Twenty-one KEGG (Kyoto Encyclopedia of Genes and Genomes) and 25 BioCarta pathways were significantly associated with DEGs of 100 μM 8-OHD treatment (Appendix A). Among these, the MAPK signal pathway was present in both KEGG and BioCarta pathways. 8-OHD-upregulated and downregulated genes involved in the KEGG MAPK signal pathway (hsa04010) were marked as red and yellow, respectively, in Figure 8a. Gene Set Enrichment Analysis (GSEA) reveals that the BioCarta MAPK pathway was predominantly correlated with upregulated DEGs of 8-OHD-treated K562 cells (Figure 8b).

Furthermore, we used the commercial MetaCore package to analyze the pathway enrichment by DEGs, and the top 15 significantly enriched pathways were shown in Figure 8c. Two MAPK-related and one oxidative stress-related signaling pathways were significantly regulated by 8-OHD. The canonical pathway map, which represents the neurogenesis Nerve growth factor/ Tropomyosin receptor kinase A (NGF/TrkA) MAPK signaling pathway in response to 8-OHD treatment, is shown in Appendix A. It has been reported that neurotrophin signaling endosomes are formed in CML cells as in neuronal cells, and imatinib treatment enhances NGF/TrkA-mediated signaling in K562 cells [77]. Current result indicates that 8-OHD enhanced NGF/TrkA MAPK signaling pathway, which may be due to its BCR-ABL lowered effect in K562 cells. Furthermore, NGF/TrkA MAPK signaling may be involved in 8-OHD-mediated K562 differentiation.

Comparative enrichment analysis of the DEG dataset by MetaCore identified top 23 process networks (Figure 8d). It was found that hemopoiesis, cell cycle regulation, NF-κB-, and MAPK-, and JAK/STAT-mediated apoptosis/anti-apoptosis networks are significantly regulated by 8-OHD. The JAK/STAT pathway is very important in transferring extracellular signals to the cell and initiates gene expression involved in cell proliferation, differentiation, survival, and developmental processes [78]. The network map of MAPK and JAK/STAT-mediated apoptosis/anti-apoptosis is shown in Appendix A. It has been reported that 8-OHD triggered apoptosis in breast cancer stem-like cells (BCSCs) through blocking of the JAK/STAT signaling pathway [79]. Here, we found that JAK/STAT-related genes, *SYK, STAT3, STAT5,* and *BIRC5,* were all downregulated by 8-OHD in K562 cells, indicating that the JAK/STAT signaling pathway was attenuated.

### 3.9. 8-OHD Modulates MAPK Pathways and NF-kB Activation

Since DAVID, Metacore, and GSEA analyses indicated the potential regulation of MAPK signaling pathways by 8-OHD, we continued to analyze MAPK activation by Western blot. It was found that 8-OHD (25–100 μM) significantly increased ERK (extracellular signal-regulated kinases), JNK (c-Jun N-terminal kinase), and p38 phosphorylation in a dose-dependent manner after 24 h treatment (Figure 9a). The activation of the ERK signaling pathway has been reportedly associated with K562 differentiation [34], while JNK activation is involved in ROS-mediated autophagy [80]. Furthermore, JNK and p38 phosphorylation have been implicated in apoptosis in K562 cells [81,82]. Here, we found that 8-OHD activated all three MAPK signaling pathways, which may be responsible for the differentiation and cell death in K562 cells.

We also found that *RelA* (NF-κB p65) mRNA expression was induced by 8-OHD (Figure 8a). Thus, we continued investigating the effect of 8-OHD on NF-κB activation by analyzing the level of phosphorylation of p65 in nuclear extracts of K562 cells. It was found that the phosphorylation of nuclear p65 increased dose-dependently 24 and 48 h after 8-OHD (25–100 μM) treatments (Figure 9b). NF-κB is a multifunctional transcription factor that mediates pro- and anti-apoptotic as well as pro-autophagic signals [83,84]. In addition, NF-κB and MAP kinase pathways are involved in PMA-induced megakaryocytic differentiation of K562 cells [85]. NF-κB is a positive regulator for p21^Cip1^ expression and induces cell cycle arrest and apoptosis [86]. Therefore, 8-OHD activated NF-κB, which may be responsible for the differentiation, cell cycle arrest, and apoptosis in K562 cells.

## 4. Conclusions

Our study provides the first evidence for the anti-CML activities and underlying molecular mechanisms of 8-OHD. Our results demonstrated that K562 cell proliferation was significantly inhibited by 8-OHD in a dose- and time-dependent manner. 8-OHD could induce K562 cell cycle arrest at the S phase by the upregulation of p21^Cip1^, a negative regulator of the cell cycle, accompanied with downregulation of cyclin D (CCND2) and cyclin D-dependent kinase (CDK6). The treatment of K562 cells with 8-OHD (100 μM) resulted in ROS overproduction, caspase-7-dependent apoptosis, autophagy, and decreased BCR-ABL. Furthermore, 8-OHD also promoted EGR1-induced K562 differentiation as megakaryocytic marker genes, *CD61* and *CD42b*, were upregulated and the formation of large cells with multi-lobulated nuclei in K562 cells was found (Figure 10).

A microarray-based transcriptome profiling was performed to offer a better understanding of the effects of 8-OHD in K562 cells. A total of 3174 differentially expressed genes (DEGs) (1522 upregulated and 1652 downregulated) were identified from K562 cells 48 h after 8-OHD (100 μM) incubation. The enrichment analysis showed that GO biological processes, such as small GTPase-mediated signal transduction, autophagy, transcription from RNA polymerase II promoter, and protein phosphorylation, were significantly associated with DEGs. Gene Set Enrichment Analysis (GSEA) reconfirms that most of the genes involved in the regulation of apoptosis were significantly stimulated by 8-OHD. Protein–protein interaction (PPI) network was constructed for upregulated DEGs associated with autophagy (GO: 0006914) using String. It was found that *p62/UVRAV*-mediated autophagy may be responsible for the decreased level of BCR-ABL.

The 8-OHD-modulated MAPK signaling pathway was confirmed using KEGG Pathview library, BioCarta and MetaCore package. The results of Western blot analysis reveal that 8-OHD (25–100 μM) significantly increased phosphorylation of ERK, JNK, and p38 MAPK in a dose-dependent manner in K562 cells. The phosphorylation of nuclear NF-κB-p65 was also increased by 8-OHD treatment dose-dependently. The combination of experimental and in silico analyses could provide a more thorough understanding about potential development of 8-OHD in CML therapy.

## Figures and Tables

**Figure 1 biomedicines-08-00506-f001:**
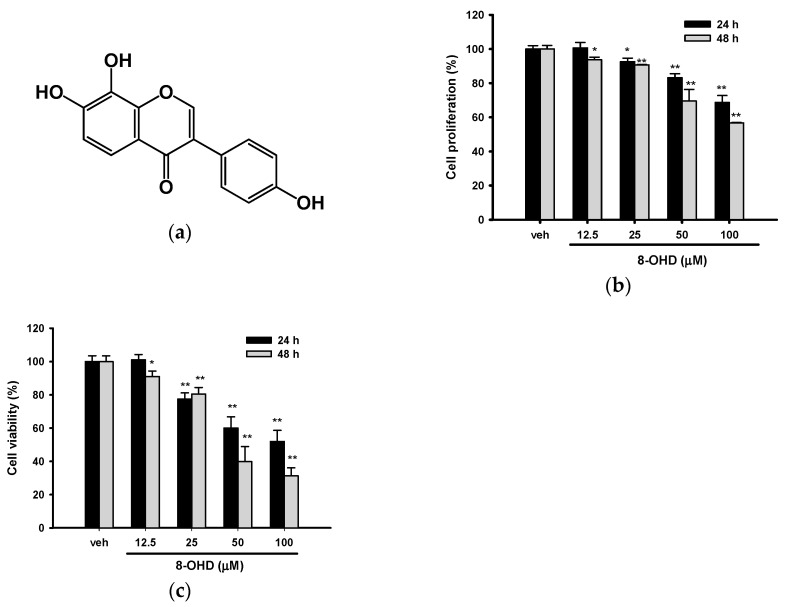
Effects of 8-hydroxydaidzein (8-OHD) on the cell viability of K562 cells. (**a**) Chemical structure of 8-OHD. K562 cells were treated with vehicle (0.1% DMSO) or 8-OHD (6.25–100 μM) for 24 and 48 h. (**b**) Cell viability was measured by 3-(4,5-dimethylthiazol-2-yl)-2,5-diphenyltetrazolium bromide (MTT) assay. (**c**) Cell viability was examined using trypan blue exclusion test. The experiments were repeated three times. These data represent the mean ± SD of three independent experiments. * *p* < 0.05 and ** *p* < 0.01 represent significant differences compared with the vehicle-treated cells.

**Figure 2 biomedicines-08-00506-f002:**
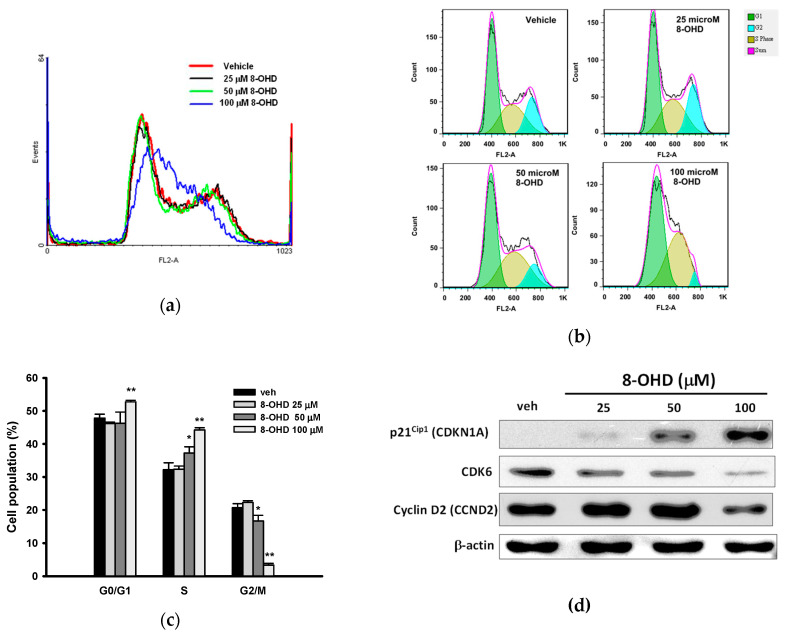
Effects of 8-OHD on the cell cycle distribution and regulatory gene expression. K562 cells were treated with vehicle or 8-OHD (25, 50, and 100 μM) for 24 h, and distribution of the cell cycle was measured by flow cytometric analysis. (**a**) FL2-A histogram overlay of K562 cells treated with various concentrations of 8-OHD. (**b**) FlowJo software analysis of DNA cell cycle. (**c**) The cell populations in G_0_/G_1_, S, and G_2_/M phases were quantified as described in Section 2.4. The experiments were replicated three times. These data represent the mean ± SD of independent experiments. * *p* < 0.05 and ** *p* < 0.01 represent significant differences compared with the vehicle-treated cells. (**d**) Western blot analysis of expression of p21^Cip1^, CDK6, cyclin D2, and β-actin of K562 cells treated with 8-OHD (25, 50, and 100 μM) for 24 h.

**Figure 3 biomedicines-08-00506-f003:**
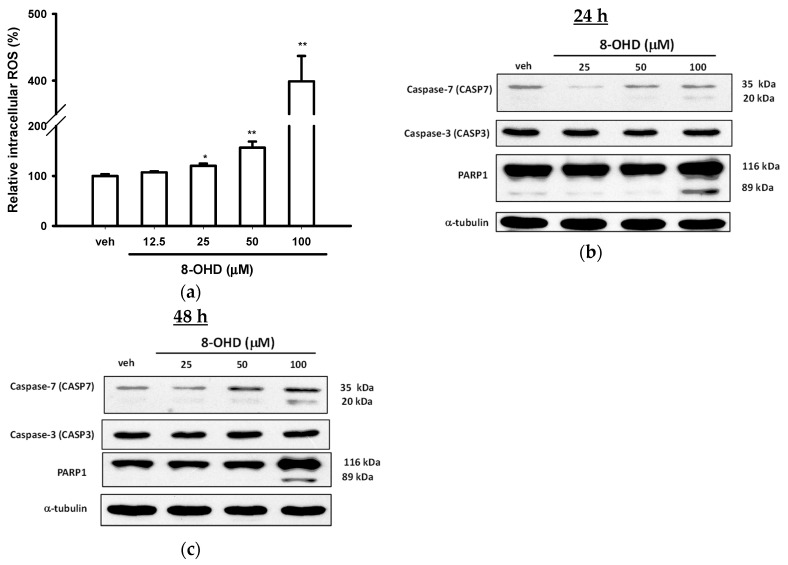
Effects of 8-OHD on reactive oxygen species (ROS) production and apoptosis in K562 cells. (**a**) K562 cells were exposed to various concentrations of 8-OHD for 24 h. Intracellular ROS was measured by DCF fluorescence as described in Materials and Methods. These data represent the mean ± SD of independent experiments. * *p* < 0.05 and ** *p* < 0.01 represent significant differences compared with the vehicle-treated cells. (**b**,**c**) Western blot analysis of apoptosis-related gene expression 24 and 48 h after 8-OHD treatment.

**Figure 4 biomedicines-08-00506-f004:**
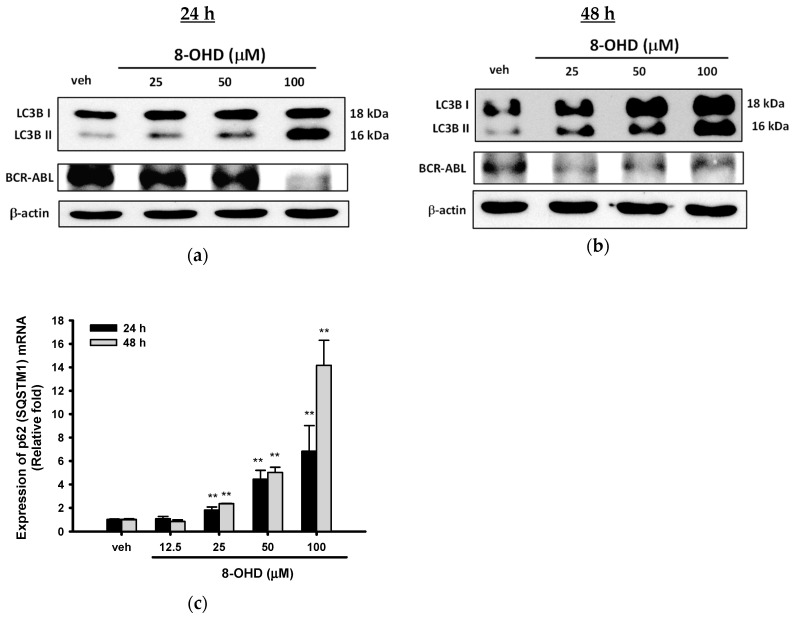
Effects of 8-OHD on autophagy and BCR-ABL protein level in K562 cells. (**a**,**b**) Western blot analysis of autophagy-related gene expression and BCR-ABL oncoprotein in K562 cells treated with 8-OHD for 24 and 48 h. (**c**) mRNA expression of p62/ SQSTM1 was analyzed by RT-Q-PCR as described in Materials and Methods. ** *p* < 0.01 represents significant differences compared with the vehicle-treated cells.

**Figure 5 biomedicines-08-00506-f005:**
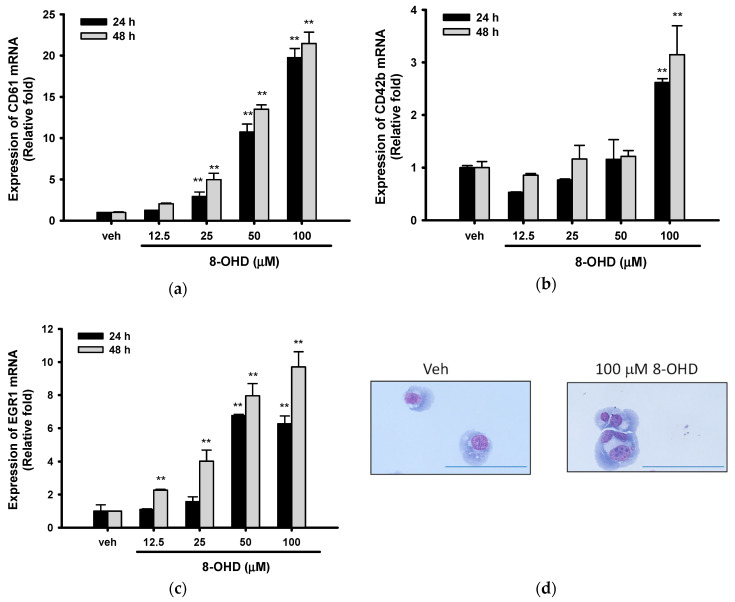
Effects of 8-OHD on the differentiation of K562 cells. (**a**–**c**) mRNA expression of CD61, CD42b, and EGR1 were analyzed by RT-Q-PCR as described in Materials and Methods. ** *p* < 0.01 represents significant differences compared with the vehicle-treated cells. (**d**) Cells were treated with vehicle (0.1% DMSO) or 8-OHD (100 μM) for 48 h. K562 cells were stained with Wright–Giemsa dye and the cell morphological changes and multi-lobulation of the nucleus were observed under microscopy. The scale bar is 100 μm.

**Figure 6 biomedicines-08-00506-f006:**
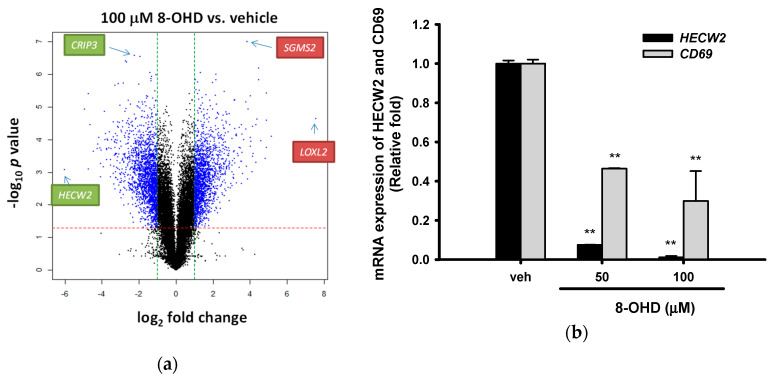
Microarray analysis of 8-OHD-treated K562 cells. (**a**) Volcano plots of total gene expression profiles of the K562 cells treated with 8-OHD (100 μM) for 48 h. Each dot represents the mean expression (n = 2) of the individual gene obtained from a microarray normalized dataset. The green and red dotted lines (cut-off values) were established according to the following parameters: fold= |2| and *p*-value = 5%, respectively. Genes above the cut-off lines have been considered as differentially expressed genes (DEG) and are shown as blue dots. (**b**) RT-Q-PCR analysis of HECW2 and CD69 expression in K562 cells 48 h after 8-OHD treatment (50 and 100 μM). ** *p* < 0.01 represents significant differences compared with the vehicle-treated cells.

**Figure 7 biomedicines-08-00506-f007:**
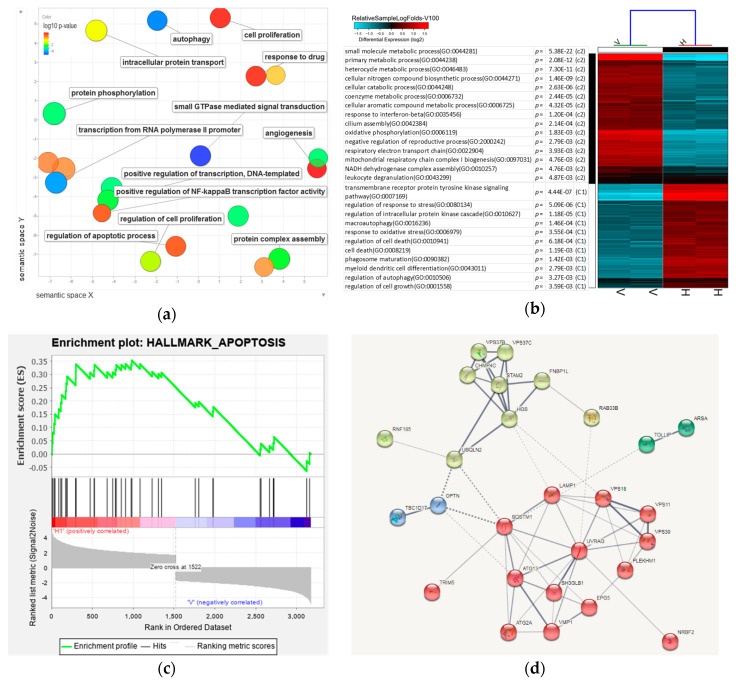
Enrichment analysis and visualization of Gene Ontology (GO), Gene Set Enrichment Analysis (GSEA), as well as predicted protein–protein interaction for DEGs in response to 8-OHD. (**a**) Significantly enriched GO BP (biological process) in 8-OHD (100 μM)- vs. vehicle (0.1% DMSO)-treated K562 cells was analyzed by REVIGO. The scatterplot represents functional clusters, the bubble color indicates the *p*-values of the GO analysis, and the bubble size indicates the frequency of the GO term in the underlying gene ontology database. (**b**) Heatmap of DEGs in response to 100 μM 8-OHD treatment. The GO listed in the upper half of the heatmap is gene ontology annotation BP of the downregulated DEGs. The lower processes are the BP of the upregulated DEGs. Only some of the representative GO BPs are shown. (**c**) Gene Set Enrichment Analysis (GSEA) demonstrates that the signature “Hallmark_Apoptosis” gene set is enriched in the DEGs of 8-OHD. The barcode plot indicates the position of the genes in each gene set. The horizontal bar in graded color from red to blue indicates positive and negative regulation by 8-OHD. The vertical axis in the lower plot indicates Ranked List Metric. (**d**) Predicted protein–protein interaction for 8-OHD-upregulated DEGs associated with autophagy (GO:0006914). The corresponding genes were uploaded to query the STRING interaction database (https://string-db.org/cgi/input.pl). Only the “Experiments”, “Databases”, and “Textmining” source options were selected, and the minimum interaction score was set to 0.4. For visual clarity, disconnected nodes were omitted from the interaction graph. Line thickness indicates the strength of data support. The identified clusters are colored differently. The solid and the dotted lines indicate connection within the same and different cluster, respectively.

**Figure 8 biomedicines-08-00506-f008:**
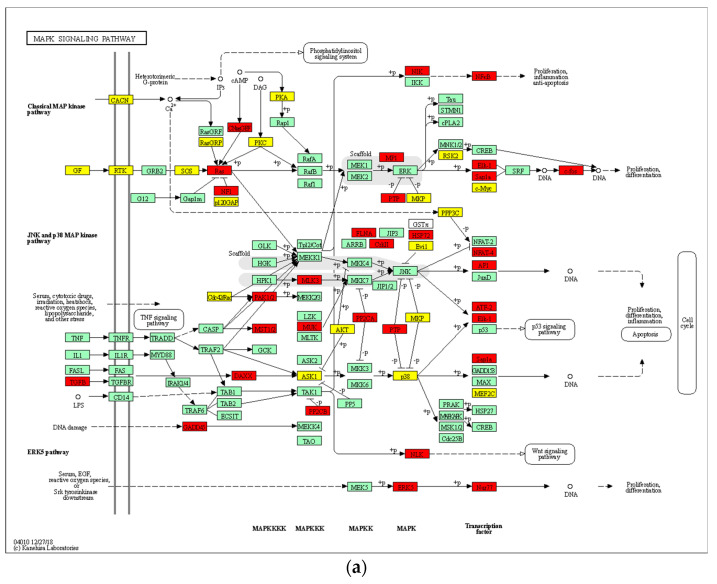
Signaling pathways and process networks associated with 8-OHD treatment. (**a**) KEGG MAPK signaling pathway (hsa04010) in K562 cells in response to 8-OHD (100 μM) treatment for 48 h. Expression changes of target genes are mapped by colors; red color: statistically significant increase in expression, yellow color: statistically significant decrease in expression, green color: expression statistically insignificant. (**b**) Gene Set Enrichment Analysis (GSEA) demonstrated that BioCarta MAPK pathway gene set was enriched in the DEGs of 8-OHD. (**c**) Top 15 significantly enriched pathways according to the MetaCore database in response to 8-OHD treatment. (**d**) Top 23 significantly enriched process networks according to the MetaCore database in response to 8-OHD treatment.

**Figure 9 biomedicines-08-00506-f009:**
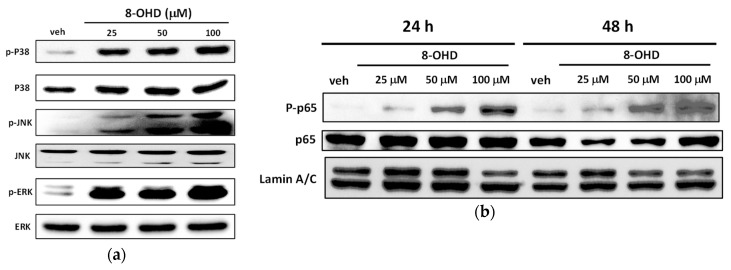
Western blotting analysis of MAPK and NF-kB activation. (**a**) Western blotting analysis of MAPK activation from total cell lysate of K562 cells treated with 8-OHD (25–100 μM) for 24 h. (**b**) Western blotting analysis of NF-κB activation from nuclear extract of K562 cells treated with 8-OHD (25–100 μM) for 24 and 48 h.

**Figure 10 biomedicines-08-00506-f010:**
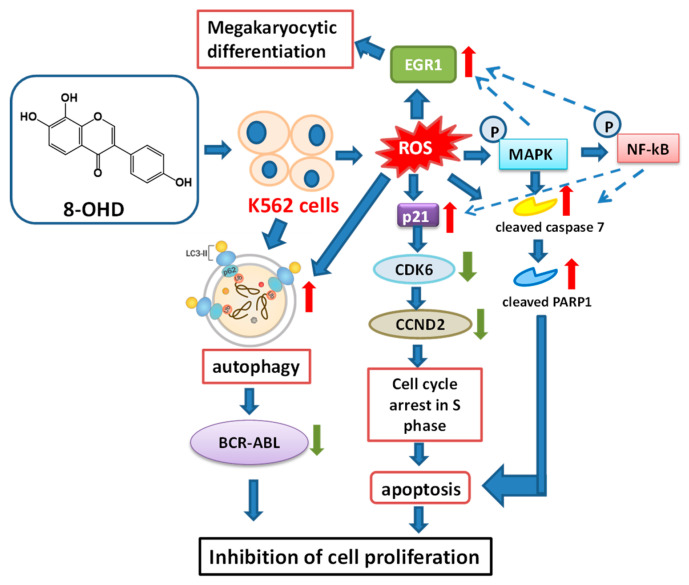
A hypothetical model for the anti-leukemic effects of 8-OHD in K562 cells. 8-OHD induces ROS overproduction, cell cycle arrest at the S phase, caspase 7-dependent apoptosis, and autophagy, as well as a decreased BCR-ABL oncoprotein level. It also promotes megakaryocytic differentiation in human CML K562 cells. MAPK and NF-κB signaling pathways may be responsible for the stimulation of apoptosis, autophagy, and differentiation in K562 cells.

**Table 1 biomedicines-08-00506-t001:** Primary antibodies used in Western blotting.

Antibody	Company	Catalog Number
α-Tubulin	Sigma	T6199
β-Actin	Genetex	GTX629630
c-ABL	Cell Signaling	2862
Caspase-3	Cell Signaling	9662
Caspase-7	Cell Signaling	9492
CDK6	Cell Signaling	13331
Cyclin D2	Cell Signaling	3741
JNK2	Cell Signaling	9258
Lamin A/C	Genetex	GTX101127
LC3B	Genetex	GTX127375
P21	Cell Signaling	2947
p38 MAPK	Cell Signaling	9212
p44/42 MAP Kinase	Cell Signaling	4695
p65 NF-κB	Cell Signaling	8242
PARP1	Santa Cruz	Sc-7150
Phoshpo-p38 MAPK	Cell Signaling	9215
Phoshpo-p44/42 MAPK	Cell Signaling	4370
Phospho-JNK1/2	Cell Signaling	4668
Phospho-p65 NF-κB	Cell Signaling	3033

**Table 2 biomedicines-08-00506-t002:** The primer pairs used in real-time PCR.

Gene	Primer Sequence (5′-3′)	Amplicon (bp)
*β-actin*	CATGTACGTTGCTATCCAGGC	250
	CTCCTTAATGTCACGCACGAT	
*ITGB3 (CD61)*	TGTATGGGACTCAAGATTGGA	187
	AGCGATGGCTATTAGGTTCA	
*GP1BA (CD42b)*	TCTGTATCAGAAGCCCTGTCTTCAC	112
	GCATCGGGAGCTTTGTCTTG	
*EGR1*	CCGCAGAGTCTTTTCCTGAC	200
	TGGGTTGGTCATGCTCACTA	
*CD69*	GCTGGACTTCAGCCCAAAATGC	121
	AGTCCAACCCAGTGTTCCTCTC	
*HECW2*	GAGTATCCGCAGAACCATGACC	133
	GGTAGAGTGAGGCAGGATGTTC	
*SQSTM1 (P62)*	AAGCCGGGTGGGAATGTTG	116
	CCTGAACAGTTATCCGACTCCAT	

**Table 3 biomedicines-08-00506-t003:** Genes associated with autophagy categories (GO: 0006914) in response to 100 μM 8-OHD.

ID	Symbol	Description	log_2_FC	*p* Value	*q* Value
8878	*SQSTM1*	autophagy receptor that interacts directly with both the cargo to become degraded and an autophagy modifier of the MAP1 LC3 family	3.48	1.79 × 10^−4^	7.76 × 10^−3^
54472	*TOLLIP*	component of the signaling pathway of IL-1 and Toll-like receptors.	2.37	7.55 × 10^−4^	1.20 × 10^−^^2^
55823	*VPS11*	vacuolar protein sorting-associated protein 11 homolog; plays a role in vesicle-mediated protein trafficking to lysosomal compartments including the endocytic membrane transport and autophagic pathway	2.3	3.22 × 10^−4^	9.01×10^−3^
10254	*STAM2*	signal transducing adapter molecule 2; Involved in intracellular signal transduction mediated by cytokines and growth factors.	2.15	1.61 × 10^−3^	1.62 × 10^−2^
3916	*LAMP1*	lysosomal associated membrane protein 1	1.98	7.97 × 10^−4^	1.21 × 10^−2^
51100	*SH3GLB1*	SH3 domain containing GRB2 like endophilin B1	1.87	5.95 × 10^−3^	3.06 × 10^−2^
81671	*VMP1*	vacuole membrane protein 1	1.8	8.26 × 10^−5^	6.26 × 10^−3^
85363	*TRIM5*	tripartite motif containing 5	1.75	1.77 × 10^−2^	6.08 × 10^−2^
23130	*ATG2A*	autophagy related 2A	1.74	1.07 × 10^−3^	1.36 × 10^−2^
91445	*RNF185*	ring finger protein 185	1.64	2.70 × 10^−4^	8.72 × 10^−3^
9146	*HGS*	hepatocyte growth factor-regulated tyrosine kinase substrate	1.62	1.18 × 10^−3^	1.42 × 10^−2^
9842	*PLEKHM1*	pleckstrin homology and RUN domain containing M1	1.61	1.70 × 10^−3^	1.67 × 10^−2^
79735	*TBC1D17*	TBC1 domain family member 17	1.57	1.47 × 10^−4^	7.34 × 10^−3^
83452	*RAB33B*	RAB33B, member RAS oncogene family	1.54	3.33 × 10^−3^	2.26 × 10^−2^
7405	*UVRAG*	UV radiation resistance associated	1.47	8.62 × 10^−3^	6.33 × 10^−3^
9776	*ATG13*	autophagy related 13	1.41	4.72 × 10^−3^	2.70 × 10^−2^
3428	*IFI16*	interferon gamma inducible protein 16	1.35	9.18 × 10^−3^	3.97 × 10^−2^
55048	*VPS37C*	VPS37C, ESCRT-I subunit	1.31	5.69 × 10^−3^	2.99 × 10^−2^
57617	*VPS18*	VPS18, CORVET/HOPS core subunit	1.31	2.87 × 10^−4^	8.81 × 10^−3^
79142	*PHF23*	PHD finger protein 23	1.3	2.16 × 10^−3^	1.85 × 10^−2^
54874	*FNBP1L*	formin binding protein 1 like	1.3	6.60 × 10^−3^	3.25 × 10^−2^
57724	*EPG5*	ectopic P-granules autophagy protein 5 homolog	1.26	1.06 × 10^−3^	1.35 × 10^−2^
79720	*VPS37B*	VPS37B, ESCRT-I subunit	1.25	3.11 × 10^−6^	3.66 × 10^−3^
29978	*UBQLN2*	ubiquilin 2	1.23	4.25 × 10^−3^	2.55 × 10^−2^
23339	*VPS39*	VPS39, HOPS complex subunit	1.23	1.29 × 10^−3^	1.47 × 10^−2^
92421	*CHMP4C*	charged multivesicular body protein 4C	1.18	1.76 × 10^−2^	6.05 × 10^−2^
410	*ARSA*	arylsulfatase A	1.18	9.04 × 10^−3^	3.94 × 10^−2^
10133	*OPTN*	optineurin	1.04	2.95 × 10^−3^	2.14 × 10^−2^
29982	*NRBF2*	nuclear receptor binding factor 2	1.01	5.32 × 10^−4^	1.07 × 10^−2^
7957	*EPM2A*	epilepsy, progressive myoclonus type 2A, Lafora disease (laforin)	−1.02	4.22 × 10^−4^	9.84 × 10^−3^
83734	*ATG10*	autophagy related 10	−1.29	8.40 × 10^−5^	6.28 × 10^−3^
51160	*VPS28*	VPS28, ESCRT-I subunit	−1.46	7.59 × 10^−3^	3.52 × 10^−2^
51028	*VPS36*	vacuolar protein sorting 36 homolog	−1.53	1.26 × 10^−2^	4.84 × 10^−2^
93343	*MVB12A*	multivesicular body subunit 12A	−1.86	1.28 × 10^−4^	7.06 × 10^−3^
84938	*ATG4C*	autophagy related 4C cysteine peptidase	−2.05	8.90 × 10^−4^	1.26 × 10^−2^

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
