# Peer review of "8-Hydroxydaidzein, an Isoflavone from Fermented Soybean, Induces Autophagy, Apoptosis, Differentiation, and Degradation of Oncoprotein BCR-ABL in K562 Cells"

_biomedicines, 2020, doi:10.3390/biomedicines8110506_

Round 1
Reviewer 1 Report
8-hydroxydaidzein (8-OHD), also known as NSC 678112, belongs to the class of organic compounds known as isoflavones. Isoflavones are naturally occurring dietary phytoestrogens distributed in the leaves, seeds, bark, and flowers of some plants, specifically in legumes, such as soybeans. The authors showed a hypothetical model for the antileukemic effects of 8-OHD in K562 cells. 8-OHD induces reactive oxygen species (ROS) overproduction, cell cycle arrest at S phase, caspase 7-dependent apoptosis, and autophagy. 8-OHD decreases BCR-ABL oncoprotein level and promotes megakaryocyte differentiation in human chronic myeloid leukemia (CML) K562 cells.
Authors declare in Abstract that this is a first report on the antileukemic effects of 8-OHD. However, the article Chen YC, Inaba M, Abe N, Hirota A. Antimutagenic activity of 8-hydroxyisoflavones and 6-hydroxydaidzein from soybean miso. Biosci Biotechnol Biochem. 2004; 68: 1372-1374 showed antiproliferative activity toward HL-60 human promyelocytic leukemia cells. This article should be added to References.
A list of abbreviations should be added. There is a mistake in reference number 40 Lo YL. ........ where a year should be 2013 and not 2012.
Reviewer 2 Report
In their manuscript “8-Hydroxydaidzein, an Isoflavone from Fermented Soybean, Induces Autophagy, Apoptosis, Differentiation, and Degradation of Oncoprotein BCR-ABL in K562 Cells” Wu et al. use a human chronic myeloid leukemia (CML) cell line (K562), treat it with soybean-purified 8-OHD, and found cell viability/prolifaration reduced, cell cycle arrest (S-phase), ROS, apoptosis, and autophagy increased. To further study relevant pathways, they looked at protein expression (Western blot), RNA-expression (qRPCR, and microarray to identify differentially expressed genes = DEGs). The authors highlight that BCR-ABL is downregulated. BCR-ABL is a fusion oncoprotein caused by reciprocal translocation between chromosomes 9 and 22 in CML. Based on their findings the authors suggest a signaling model in which 8-OHD inhibits proliferation of K562 by increasing apoptosis, autophagy, and megakaryocytic differentiation through ROS and downstream mediators/targets such as MAPK, NF-kb, caspase 7, p21, CDK, BCR-ABL. The manuscript is well written, the data look convincing and will be of interest in the field. I only have minor concerns to address, that are not asking for additional experiments as necessity.
Major point:
- 8-OHD is reported in several studies (including studies by the same authors) as antioxidant, how does it increase ROS in K562, then? Please discuss how 8-OHD might regulate ROS in opposite directions in different cell types.
Minor concerns:
- Figure 2: make sure to use the same scale bar in panels a-d. It would also be helpful to illustrate the four graphs in one plot to see the differences in the curves. This could be done with an additional panel.
- The authors include a KEGG pathway in figure 8. Please make sure to appropriately cite the KEGG pathway as the creators ask for: https://www.genome.jp/kegg/kegg1.html
- Quantification listed under 3.5 (lines 92-94). How many experiments did you do? What is the error and p-value for the differences?
- Microarray: how many genes were included in the hybridization array?
- Other cancer studies report decreased proliferation, too. Please discuss and put in perspective with your results. E.g. Tai et al., 2009/ Goh et al., 2012 (melanoma) or Rajabi et la., 2020 (breast cancer), the latter also reports JAK/STAT signaling involved.
